# Cytoreductive Surgery with Hyperthermic Intraperitoneal Chemotherapy for Gastric Cancer with Peritoneal Carcinomatosis: Additional Information Helps to Optimize Patient Selection before Surgery

**DOI:** 10.3390/cancers15072089

**Published:** 2023-03-31

**Authors:** Hao-Chien Hung, Po-Jung Hsu, Chao-Wei Lee, Jun-Te Hsu, Ting-Jung Wu

**Affiliations:** Department of General Surgery, Chang-Gung Memorial Hospital at Linkou, Chang-Gung University College of Medicine, Taoyuan 33305, Taiwanhsujt2813@cgmh.org.tw (J.-T.H.)

**Keywords:** gastric cancer, peritoneal carcinomatosis, cytoreductive surgery, hyperthermic intraperitoneal chemotherapy

## Abstract

**Simple Summary:**

Gastric cancer-associated peritoneal carcinomatosis (GCPC) is a devastating disease, and the median life expectancy is short without effective treatment. Increasing evidence shows that a combination of cytoreduction surgery (CRS) and hyperthermic intraperitoneal chemotherapy (HIPEC) for GCPC have survival benefits for certain patients. An ideal preoperative patient selection criterion for the CRS-HIPEC operation has yet to be established. This study identified two easily measurable preoperative clinical factors, the number of computed tomography (CT) prognostic risks and serum neutrophil-to-lymphocyte ratio (NLR), to predict overall survival (OS) in patients with GCPC after the CRS-HIPEC operation.

**Abstract:**

(1) Background: The prognosis of gastric cancer-associated peritoneal carcinomatosis (GCPC) is poor, with a median survival time of less than six months, and current systemic chemotherapy, including targeted therapy, is ineffective. Despite growing evidence that cytoreductive surgery with hyperthermic intraperitoneal chemotherapy (CRS-HIPEC) for GCPC improves overall survival (OS), optimal patient selection remains unclear. We aimed to evaluate preoperative clinical factors and identify indicative factors for predicting postoperative OS in patients with GCPC undergoing CRS-HIPEC. (2) Methods: We retrospectively reviewed 44 consecutive patients with GCPC who underwent CRS-HIPEC between May 2015 and May 2021. Data on demographics and radiologic assessment were collected and analyzed. (3) Results: Elevated preoperative serum neutrophil-to-lymphocyte ratio > 4.4 (*p* = 0.003, HR = 3.70, 95% CI = 1.55–8.79) and number of computed tomography risks > 2 (*p* = 0.005, HR = 3.26, 95% CI = 1.33–7.98) were independently indicative of OS post-surgery. A strong correlation was observed between intraoperative peritoneal cancer index score and number of computed tomography risks (*r* = 0.534, *p* < 0.0001). Two patients after CRS-HIPEC ultimately achieved disease-free survival for more than 50 months. (4) Conclusions: Our experience optimizes GCPC patients’ selection for CRS-HIPEC, may help to improve outcomes in the corresponding population, and prevent futile surgery in inappropriate patients.

## 1. Introduction

Up to 40% of patients with gastric cancer (GC) are diagnosed at an advanced stage [1,2,3]. Disease recurrence after curative-intent surgery is common in nearly 30% of patients with primary GC. Moreover, one-third of patients with recurrent disease develop multiple sites of metastases, and one-fifth of these patients present with a peritoneal lesion [4]. Overall, peritoneal carcinomatosis (PC) accounts for 35% of all synchronous metastases at the time of initial diagnosis [1,2,3]. A recent study even revealed a rapid increment in the proportion of distant gastric cancer cases up to 44.7% in the year 2019 [5]. The prognosis for gastric cancer-associated peritoneal carcinomatosis (GCPC) remains poor, with a median life expectancy of ≤6 months [2,6].

Despite the current systemic chemotherapy and targeted therapy demonstrating survival advantages compared with the best supportive care, the median survival time for patients with GCPC receiving such treatments is approximately 8–14 months, and long-term survival is virtually impossible [7,8]. It reflects the fact that systemic therapy alone may not be sufficient to treat peritoneal metastasis. It is hypothetical that the “peritoneal–blood barrier” reduces the drug penetration into the peritoneal cavity and therefore declines the effectiveness of systemic therapy in patients with GCPC. [9] As a potential treatment option, the benefits of a combination of cytoreduction surgery (CRS) and hyperthermic intraperitoneal chemotherapy (HIPEC) for GCPC have been demonstrated in several studies [10,11,12]. Additionally, a meta-analysis reported that the CRS-HIPEC operation improves survival in patients with GCPC without increasing complications compared to CRS alone [13].

To date, an ideal preoperative patient selection criterion for CRS-HIPEC has not been established to prevent futile surgery in patients with GCPC. In clinical practice, there is a need to identify the potential beneficiaries of the CRS-HIPEC operation by which complete cytoreduction is often accompanied by complex surgery including multi-organ resection. The hazard of resulting surgical complications may postpone scheduled systemic therapy. In selective GCPC patients, the potentially curative role of the CRS-HIPEC operation has been continually evolving and this may change future treatment guidelines in the coming years. This study aimed to evaluate preoperative clinical factors and identify indicative factors for predicting postoperative overall survival (OS) in patients with GCPC undergoing the CRS-HIPEC operation to optimize patient care.

## 2. Materials and Methods

### 2.1. Study Population

The retrospective review of consecutive CRS-HIPEC procedures for pathologically proven GC was conducted at the Linkou Chang-Gung Memorial Hospital between May 2015 and May 2021. All eligible patients were deemed to be medically fit for the CRS-HIPEC procedures as follows: Eastern Cooperative Oncology Group (ECOG) performance ≤ 2 and absence of extra-abdominal metastasis. GC patients without PC who had undergone a prophylactic CRS-HIPEC procedure were excluded. This study was approved by our Institutional Review Board (IRB No. 202101798B0C101).

### 2.2. Procedure Protocol and Patient Surveillance

Computed tomography prognostic risk factors for unresectable disease (CT risks) were defined according to the previous study as follows [14]: the presence of small bowel serosal or mesenteric disease, gross ascites, presence of a peritoneal lesion measuring > 5 cm, omental cake, small or large bowel obstruction, peri-hepatic nodules, ureteric obstruction, and biliary obstruction. The surgeon routinely assessed the peritoneal cancer index (PCI) score at the beginning, and the completeness of cytoreduction (CCR) degree at the end of CRS-HIPEC procedure as Sugarbaker described [15].

Peritoneal malignancy was investigated using the peritoneal cancer index (PCI), wherein Grade I presented a score of <9, II for 10–19, III for 20–29, and IV for 30–39 following the Sugarbaker classification. The PCI score is a scoring system to measure the severity and extension of peritoneal surface malignancy based on divided intraperitoneal areas; point scores at each area are then added up. The 4 classes of CCR were described as the following: CCR 0, no macroscopic residual tumor; CCR 1, the residual tumor size is less than 2.5 mm; CCR 2, the residual tumor size is between 2.5 mm and 2.5 cm; CCR3, the residual tumor size is greater than 2.5 cm. A complete cytoreduction is defined as CCR 0 to CCR 1.

There is no strict protocol of prophylactic HIPEC for non-metastatic GC currently in our institute, and neoadjuvant intraperitoneal systemic chemotherapy (NIPS) for selected patients with evidence of presenting PC is also mainly introduced by physicians’ preference. Briefly, prophylactic HIPEC after curative gastrectomy has been considered an option as an adjuvant treatment to prevent peritoneal seeding or recurrence for high-risk patients (whose tumor invaded the visceral peritoneum of the stomach or the tumor directly invaded adjacent structures) as a previous study suggested [12]. NIPS is administered laparoscopically for selected advanced GC with evident PC, and the following intraoperative findings are recorded: the PCI score, PC location, and ascites amount. Intraoperative fluid cytology and biopsy are arranged according to clinical needs. The efficacy is evaluated after a full cycle of NIPS treatment (4 times), and early termination of NIPS is considered for conditions including rapid growing or newly developed PC lesions, or intractable post-procedure complications. A plateau response of GCPC to NIPS could lead to either curative- or palliative-intent CRS-HIPEC based on an individual’s clinical progression. Furthermore, we do not perform routine preoperative laparoscopic tumor staging and PCI evaluation before CRS-HIPEC. 

Other collected data, CRS-HIPEC techniques, and post-procedure intensive care unit admission indications were considered as per standard procedures described in our previously published study [16]. The protocolized HIPEC regimens for GCPC were docetaxel 35 mg/m^2^ plus cisplatin 50 mg/m^2^ at 42–43 °C or mitomycin-C 30–50 mg/m^2^ plus cisplatin 50–100 mg/m^2^ for 90-min circulating time at 42–43 °C. The 3-week NIPS regimens were oral S-1 60 mg/m^2^ for 14 days followed by seven rest days and docetaxel 30 mg/m^2^ plus cisplatin 30 mg/m^2^ via intraperitoneal (I.P.) route on day one and intravenous route (I.V.) on day eight. After the CRS-HIPEC operation, patients adhered to institutional standardized GC follow-up, including continuing either adjuvant or palliative chemotherapy after recovery from the CRS-HIPEC operation and demand contrast-enhanced chest–abdomen–pelvis CT evaluation every three to six months and endoscopy examination at least one time every year for post-surgical oncological surveillance. The systemic therapy was given by physicians’ preference and consisted of protocol follow-up.

### 2.3. Data Collection, Data Forms, and Statistical Analysis

A standard data form was created to collect clinicopathologic information on the GCPC tumor, on the patient demographics including underlying medical history, on the previous treatment history, on the clinical symptoms, and on the surgical details. Pearson’s chi-square test and independent t-test were used to compare parameters between groups. Means ± standard deviations are used to represent continuous variables and numbers with percentages are used for categorical data. Receiver operating characteristic (ROC) curve analyses were used to examine predictive values of clinical factors by the Youden index and to determine optimal cut-off points of numbers of CT risks and neutrophil-to-lymphocyte ratio (NLR). Post-procedural OS was the primary endpoint, measured from the date of CRS-HIPEC until death or the most recent follow-up. A Cox proportional hazards regression model was used to investigate preoperative prognostic variables. Kaplan–Meier analysis with a log-rank test was used to assess survival outcomes. The statistical significance was defined as a two-tailed *p*-value of < 0.05. Statistical analyses were performed using SPSS Statistics (version 24.0; SPSS Inc., Chicago, IL, USA).

## 3. Results

### 3.1. Population Composition

A total of 73 consecutive CRS-HIPEC procedures for pathologically proven GC were reviewed. Patients with non-metastatic status who underwent prophylactic CRS-HIPEC operation (*n* = 23) and those who underwent repetitive CRS-HIPEC procedures (*n* = 6) were excluded. Subsequently, 44 GCPC cases were included in the study population, and 6 received neoadjuvant and intraperitoneal systemic chemotherapy (NIPS) before the formal CRS-HIPEC operation. All eligible patients were deemed to be medically fit for CRS-HIPEC procedures as follows: Eastern Cooperative Oncology Group (ECOG) performance ≤2 and absence of extra-abdominal metastasis. We categorized enrolled patients into the curative-intent (*n* = 20; as a therapeutic measure in those who achieved CCR 0–1) and the palliative-intent group (*n* = 24; as symptomatic mitigation in those who attained CCR 2–3) according to the CRS completeness, as shown in Figure 1.

Table 1 summarizes the demographic and clinical information of the 44 patients with GCPC who underwent CRS-HIPEC. A total of 20 (45.5%) patients received complete CRS (CCR 0–1, referred to as curative-intent), while the other 24 (54.5%) received incomplete CRS (CCR 2–3, referred to as palliative-intent). Baseline characteristics, such as age, sex, performance status, underlying comorbidities, and chemotherapy history, were not statistically different between the two groups. As expected, the number of CT prognostic risks, percentages of severe clinical symptoms, and recurrent GC status were significantly higher in the palliative-intent group (all *p*-values < 0.05) than in the curative-intent group. The number of CT prognostic risks and GC primacy were discriminative in the curative-intent group. We had 8 (40%) and 17 (70.8%) patients with more than 2 CT risks in the curative- and palliative-intent groups, respectively (*p* = 0.040). We further investigated the relationship between the peritoneal cancer index (PCI) score and these two factors. There was a strong correlation between PCI score and CT risks (r = 0.534, *p* = 0.0001), whereas GC primacy did not correlate with PCI score (*r* = 0.153, *p* = 0.320), as shown in Figure 2.

### 3.2. CRS-HIPEC Operation Intents and Outcomes

Despite the lower PCI score (11.0 ± 7.0 vs. 23.5 ± 8.1, *p* < 0.001) and neutrophil-to-lymphocyte ratio (NLR) (3.6 ± 3.2 vs. 12.8 ± 19.7, *p* < 0.001) detected in the curative-intent group, the number of organs that were resected was significantly higher than in the palliative-intent group (5.7 ± 2.9 vs. 1.5 ± 1.8, *p* < 0.001). NIPS, intraoperative blood transfusion, HIPEC procedure duration, temperature settings, and tumor histologic differentiation were similar between the two groups, as shown in Table 2. The cumulative OS survival rates in the first and third years after operation were 37.2% and 21.7%, respectively (Figure 3A). As compared to patients who received palliative-intent HIPEC, patients who received curative-intent HIPEC had better outcomes (*p* = 0.001, log-rank; Figure 3B) and were more likely to achieve long-term survival (78.7% vs. 0.0%, *p* < 0.001). A total of 2 patients in the curative-intent group eventually achieved disease-free survival over 50 months after CRS-HIPEC, and their PCI scores were 3 and 10.

### 3.3. Univariate and Multivariate Analyses of Preoperative Survival Predictors

Cox regression analysis was conducted to identify independent preoperative risk factors for survival prediction (Table 3). In the univariate analysis, several variables were identified as prognostic factors, including a ECOG score = 2, a high preoperative serum NLR > 4.4, poor histologic differentiation, and number of CT risks >2. In the multivariate analysis, the 2 identified prognostic factors were high preoperative serum NLR >4.4 (*p* = 0.003, HR = 3.70, 95% CI = 1.55–8.79) and a number of CT risks >2 (*p* = 0.005, HR = 3.26, 95% CI = 1.33–7.98). Concerning the impact of independent preoperative risks on survival, patients with a high preoperative serum NLR tended to have worse outcomes in the progressive process (*p* = 0.003; Figure 4A). In addition, survival in patients with >2 CT risks during the preoperative examination was significantly lower than that in patients with ≤2 CT risks (*p* = 0.011; Figure 4B).

### 3.4. Additional Preoperative Information to the PCI in Clinical Outcomes

The clinical significance of the PCI score in peritoneal surface malignancy patients contributing to clinical outcomes is evident but may not be always available preoperatively. Therefore, we used ROC analyses to examine predictive values of the emerging combination of the two survival risks, identified from the Cox regression model assessing purely preoperative factors and PCI class, as shown in Figure 5. The patients were categorized into 3 groups by the number of independent survival risks a patient had (14 patients had 0 risk factors, 21 patients had at least 1 risk factor, and 9 patients had both risk factors). There were 10, 12, 16, and 6 patients in the PCI classes I, II, III, and IV, respectively. The area under the ROC (AUROC) of the emerging numbers of survival risks in predicting the 6-month mortality, the 12-month mortality, and incomplete cytoreduction were 0.745 (95% CI: 0.597–0.893; Figure 5A), 0.737 (95% CI: 0.588–0.885; Figure 5B), and 0.632 (95% CI: 0.466–0.797; Figure 5C), respectively. On the contrary, predictive values of PCI class in the 6-month mortality, the 12-month mortality, and incomplete cytoreduction were 0.709 (95% CI: 0.556–0.863; Figure 5D), 0.675 (95% CI: 0.508–0.841; Figure 5E), and 0.786 (95% CI: 0.642–0.930; Figure 5F), respectively. Although the PCI class showed a good result to predict incomplete cytoreduction, the discrimination ability was mediocre in prognosticating the 6-month and the 12-month survival outcomes. In addition, the emerging preoperative risks scoring seemed to provide valuable information with regards to survival outcomes.

## 4. Discussion

This research aims to try to identify the potential GCPC beneficiaries of the CRS-HIPEC operation by analyzing the relationship between the clinicopathological information and survival outcomes for GCPC after CRS-HIPEC. In this present study, the 3-year survival rate in the curative-intent group was 38.6%. Moreover, relapse-free survival (RFS) was also assessed in patients who achieved CCR 0 (*n* = 12), and the median RFS was 18.2 months (mean ± SD: 20.7 ± 5.2; 95% CI: 10.4–30.9). Our patients with PCI class I had a median OS of 28.2 months (OS at 1-, 2-, and 3-year of 88.9%, 74.1%, and 37.0%), while those with PCI classes II, III, and IV had median OSs of 10.0, 5.7, and 2.2 months, respectively. In Manzanedo’s series with a PCI cut-off point of 7, their patients with PCI < 7 had a median OS of 26.1 months [17]. Notably, 2 GCPC patients with PCI scores of 3 and 10, attained long-term disease-free survival of over 50 months after CRS-HIPEC in our series. To date, there is still no current consensus on an optimal PCI score cut-off value for patients with GCPC to receive CRS-HIPEC [17,18]. A higher PCI score is considered to be correlated with a heavier tumor burden, which increases the difficulty in achieving complete cytoreduction. When a PCI score exceeded 13, the rate of completeness of cytoreduction was only about 7% [19], and we discovered that only 7 of 29 (24.1%) patients achieved CCR 0–1 in the present study. PCI assessment is certainly crucial for patients with GCPC receiving the CRS-HIPEC operation; however, this information is only obtained during surgery. We demonstrated a strong correlation between the number of CT risks and PCI scores, leading to distinctive outcomes. We believe that the number of CT risks is a more beneficial prognostic factor than the image-predicted PCI score because, when compared to the actual PCI score, the predicted PCI score is often underestimated. This makes it challenging to detect lesions less than 5 mm, especially in patients with GCPC who tend to present with a large amount of ascites [20,21]. The additional benefit of counting CT risks is that it is an objective, fast, and simple procedure, and it requires less interpretation expertise.

Previous studies had found that synchronous PC, good tolerance of multiple cycles (>6 times) of systemic chemotherapy, and histology other than signet ring cells were favorable predictors of better survival outcomes after CRS-HIPEC [22,23]. In addition, among the findings of this study was the significant role played by preoperative NLR in determining outcomes after CRS-HIPEC. A previous study revealed its application in dichotomizing the patients into the high- and the low-risk groups before each chemotherapy line for GC and success in predicting post-treatment outcomes [24]. Another study constructed a prediction model constituting NLR and CA19-9 in GC patients [25]. This finding is similar to our previous research on CRS-HIPEC for various cancers [16]. NLR appears to be a prominent factor in survival analysis, and we believe that more aggressive and progressive disease characteristics are reflected in tumor microenvironments. As an indicative prognostic factor, high serum NLR levels represent multifaceted evaluations of cancer patients, consisting of oncologic burden, tumor invasiveness, upregulation of inflammatory cancer-associated cytokines, decline of host anti-cancer immunity, and systemic-inflammation-associated malnutritional status [23,24,26]. In short, preoperative serum NLR is known to be associated not only with tumor characteristics, but also with patient attributes, including nutritional, functional, and immunological aspects [27,28,29]. However, it has drawn little attention in peritoneal surface malignancies regardless of being a well-studied prognostic factor in various cancers [30,31,32].

Despite current systemic chemotherapy and targeted therapy demonstrating survival advantage for GCPC treatment, it is almost impossible to achieve long-term survival. Currently, little is known about how to determine the optimal treatment for individual GCPC patients. Nevertheless, to select the optimal patients for satisfactory outcomes remains a significant challenge. It again gives particular importance to selecting optimal patients as being the best or most beneficial for receiving CRS-HIPEC. In the present study, we identified 2 independent factors (high preoperative serum NLR ≤ 4.4 and number of CT risks ≤ 2) that may serve as selection indicators for patients with GCPC to undergo CRS-HIPEC.

This paper sets out a vision of how to evaluate GCPC patients before, and to predict the prognosis after CRS-HIPEC operation. Despite these findings, this study has clear limitations. Mainly, this was a retrospective review with a relatively small number of patients enrolled at a single institution. Although it is an indisputable fact that large and heterogeneous distribution of individual survival differences exist (the minimum and maximum of the postoperative survival periods were 0.5 and 51.2 months), it provided how the degree of peritoneal cancer extent and the completeness of cytoreduction affects surgical and survival outcomes. Several prospective studies with a larger number of multi-institutional cases are needed to assess the generalizability of our results. The optimal approach for CRS-HIPEC in patients with GCPC remains unanswered. Prior to CRS-HIPEC, efforts should be made to optimize patient selection in order to improve clinical outcomes.

## 5. Conclusions

In the current study, we found 2 easily measurable and promising preoperative risk factors (number of CT prognostic risks > 2 and serum NLR > 4.4) that can be used to select ideal candidates for CRS-HIPEC operation and to predict post-surgical survival benefits. Our findings not only provide essential values, especially for those who lack or have difficulty in assessing the exact PCI score, but also give information in addition to the PCI score. Looking forward to the future, these results should inspire developments in optimizing GCPC patient selection prior to the CRS-HIPEC operation.

## Figures and Tables

**Figure 1 cancers-15-02089-f001:**
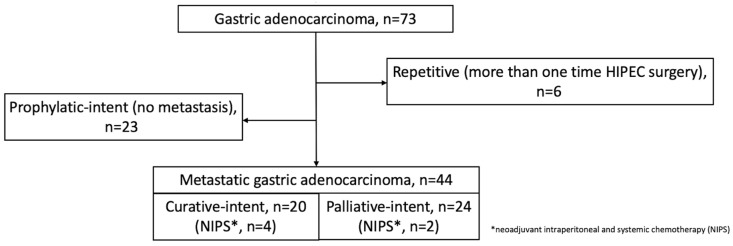
The flow diagram of the current study. Consecutive gastric cancer-associated peritoneal carcinomatosis (GCPC) patients in the institutional cytoreductive surgery with hyperthermic intraperitoneal chemotherapy (CRS-HIPEC) database during 2015 to 2021 were reviewed (*n* = 73). Exclusion was made for those who were non-metastatic GC patients receiving prophylactic HIPEC procedure (*n* = 23) and those who received repetitive CRS-HIPEC operations (*n* = 6). A total of 44 patients of CRS-HIPEC procedures, with 20 cases in the curative-intent group and the other 24 cases in the palliative-intent group, were enrolled for further overall survival (OS) analysis.

**Figure 2 cancers-15-02089-f002:**
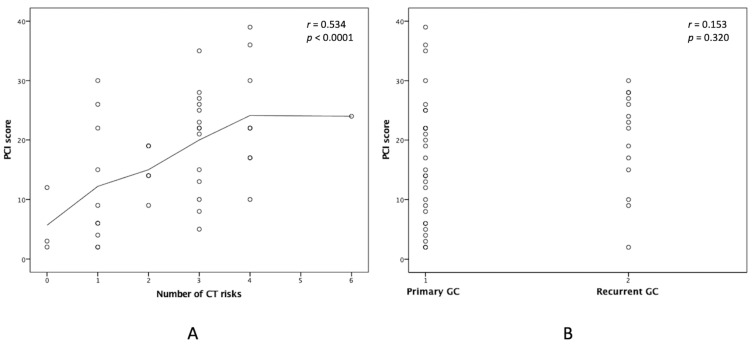
Relationships between PCI score and clinical factors. Linear regression analyses were performed, which showed (**A**) a high correlation between PCI score and number of CT prognostic risks (*r* = 0.534, *p* < 0.0001) and (**B**) an irrelevant correlation between PCI score and GC primacy (r = 0.153, *p* = 0.320). PCI, peritoneal cancer index; CT, computed tomography; GC, gastric cancer.

**Figure 3 cancers-15-02089-f003:**
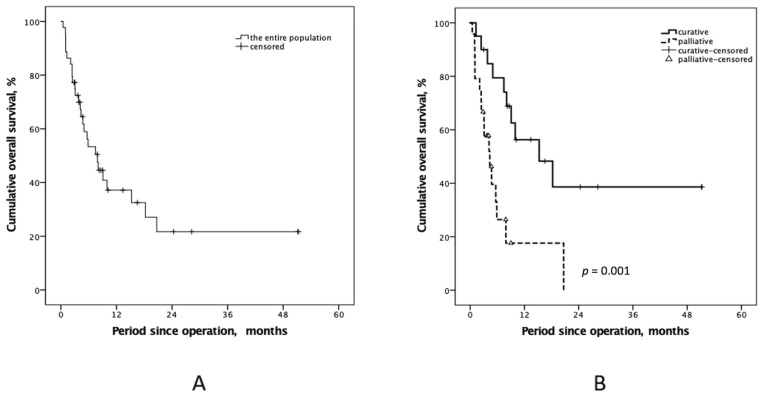
Kaplan–Meier plots of overall survival (OS) rate according to (**A**) the entire study population and (**B**) by curative- and palliative-intent. The OS rate of the curative-intent group was significantly superior, compared with the palliative group (*p* = 0.001).

**Figure 4 cancers-15-02089-f004:**
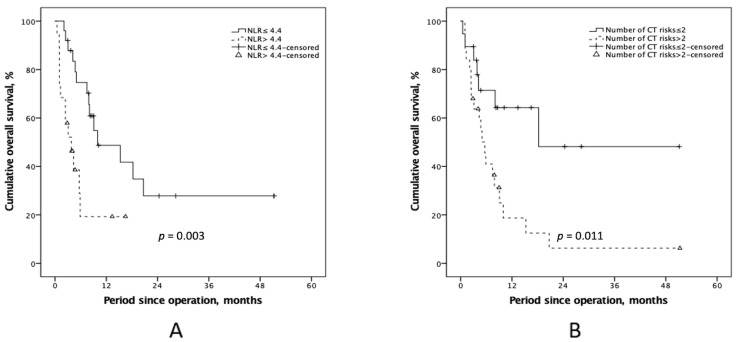
Kaplan–Meier plots of overall survival according to different subgroups that contain with or without the identified survival risks. (**A**) Patients with lower preoperative serum NLR (≤4.4) demonstrated superior outcomes (*p* = 0.003), and (**B**) a more favorable outcome after CRS-HIPEC was deliberated in patients with ≥2 CT prognostic risks (*p* = 0.011). NLR, neutrophil-to-lymphocyte ratio; CRS-HIPEC, cytoreductive surgery and hyperthermic intraperitoneal chemotherapy; CT, computed tomography.

**Figure 5 cancers-15-02089-f005:**
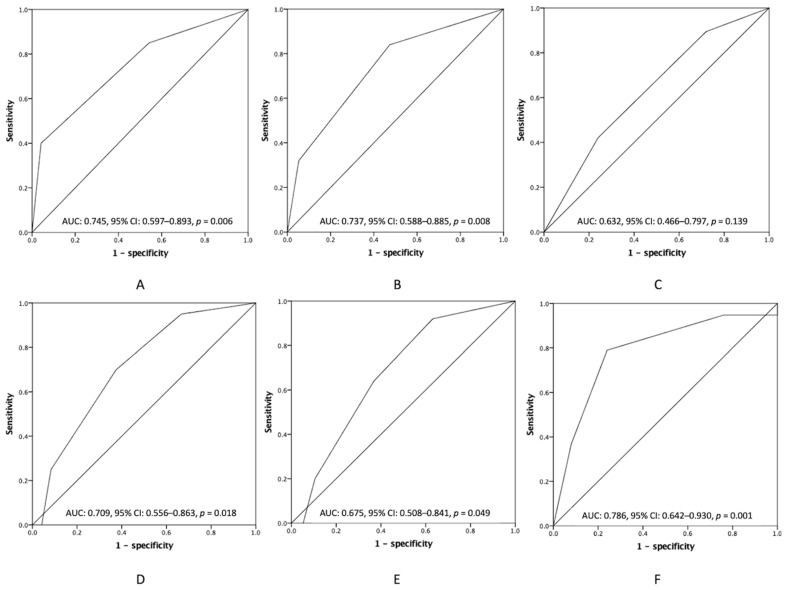
ROC curve analyses were used to examine predictive values of (**A**) postoperative 6-month mortality, (**B**) postoperative 12-month mortality, and (**C**) incomplete cytoreduction of the combination of an emerging number of CT risks and NLR, and predictive values of (**D**) postoperative 6-month mortality, (**E**) postoperative 12-month mortality, and (**F**) incomplete cytoreduction of the PCI class for GCPC patients receiving CRS-HIPEC. ROC, Receiver operating characteristic; CT, computed tomography; NLR, neutrophil-to-lymphocyte ratio; PCI, peritoneal cancer index; GCPC, gastric cancer-associated peritoneal carcinomatosis; CRS-HIPEC, cytoreductive surgery and hyperthermic intraperitoneal chemotherapy.

**Table 1 cancers-15-02089-t001:** Demographic and clinical characteristics of the metastatic gastric cancer patients who underwent CRS/HIPEC.

	Curative-Intent, *n* = 20	Palliative-Intent, *n* = 24	*p*-Value
Basic conditions			
Age, years old (≤65/>65)	19 (95.0%)/1 (5.0%)	19 (79.2%)/5 (20.8%)	0.128
Gender (Male/Female)	8 (40.0%)/12 (60.0%)	10 (41.7%)/14 (58.3%)	0.911
ECOG performance (0–1/2)	17 (85.0%)/3 (15.0%)	20 (83.3%)/4 (16.7%)	0.880
Co-morbidities/Histories			
Smoke (No/Yes)	17 (85.0%)/3 (15.0%)	21 (87.5%)/3 (12.5%)	0.810
Alcohol use (No/Yes)	18 (90.0%)/2 (10.0%)	19 (79.2%)/5 (20.8%)	0.328
Diabetes (No/Yes)	17 (85.0%)/3 (15.0%)	20 (83.3%)/4 (16.7%)	0.880
Hypertension (No/Yes)	17 (85.0%)/3 (15.0%)	18 (75.0%)/6 (25.0%)	0.413
Viral hepatitis (No/Yes)	19 (95.0%)/1 (5.0%)	21 (87.5%)/3 (12.5%)	0.389
Co-malignancy (No/Yes)	18 (90.0%)/2 (10.0%)	21 (87.5%)/3 (12.5%)	0.795
Abdomen op Hx (No/Yes)	13 (65.0%)7 (35.0%)	12 (50.0%)/12 (50.0%)	0.317
Previous C/T (No/Yes)	10 (50.0%)/10 (50.0%)	11 (45.8%)/13 (54.2%)	0.783
Ascites (No/Yes)	12 (60.0%)/8 (40.0%)	11 (45.8%)/13 (54.2%)	0.349
Number of CT risks (≤2/>2)	12 (60.0%)/8 (40.0%)	7 (29.2%)/17 (70.8%)	0.040
Clinical symptoms (None/Mild/Severe)	1 (5.0%)/18 (90.0%)/1 (5.0%)	0 (0.0%)/16 (66.6%)/8 (33.3%)	0.044
Status (Primary/Recurrent)	17 (85.0%)/3 (15.0%)	13 (54.2%)/11 (45.8%)	0.029

Abbreviation: ECOG, Eastern Cooperative Oncology Group; op, operation; hx, history; C/T, chemotherapy; CT, computed tomography.

**Table 2 cancers-15-02089-t002:** Surgical procedures, pathologic data, and outcomes of metastatic gastric cancer patients who underwent CRS/HIPEC.

	Curative-Intent, *n* = 20 ^b^	Palliative-Intent, *n* = 24	*p*-Value
Surgical characteristics			
NIPS (No/Yes)	16 (80.0%)/4 (20.0%)	22 (91.7%)/2 (8.3%)	0.261
Number of organs resected	5.7 ± 2.9	1.5 ± 1.8	<0.001
Blood transfusion (No/Yes)	14 (70.0%)/6 (30.0%)	16 (66.6%)/8 (33.3%)	0.878
HIPEC duration, mins	105.9 ± 17.1	102.8 ± 19.7	0.612
Inlet temperature, °C	43.7 ± 0.9	43.9 ± 0.9	0.476
Outlet temperature, °C	41.9 ± 0.6	39.8 ± 0.9	0.375
Highest intraoperative BT, °C	38.6 ± 0.6	39.2 ± 1.5	0.180
Tumor characteristics			
PCI score PCI class (I/II/III/IV)	11.0 ± 7.0 8 (40.0%)/8 (40.0%)/ 4 (20.0%)/0 (0.0%)	23.5 ± 8.1 2 (8.3%)/4 (16.7%)/ 12 (50.0%)/6 (25.0%)	<0.001 0.002
Differentiation (Well/ Moderate/Poor) ^a^	2 (10.0%)/1 (5.0%)/17 (85.0%)	2 (8.3%)/1 (4.2%)/21 (87.5%)	0.971
NLR	3.6 ± 3.2	12.8 ± 19.7	<0.001
NLR (≤4.4/>4.4)	15 (75.0%)/5 (25.0%)	10 (41.7%)/14 (58.3%)	0.026
Survival outcomes			
Months after diagnosing M1 status	17.2 ± 13.3	4.4 ± 4.1	0.023
OS after, months (min-max.)	14.7 ± 14.3 (1.3–51.2)	4.4 ± 4.1 (0.5–20.7)	0.001
6-month-OS rate	79.4%	26.4%	<0.001
12-month-OS rate	56.3%	17.6%	<0.001
36-month-OS rate	38.6%	0.0%	<0.001

Abbreviation: NIPS, neoadjuvant intraperitoneal and systemic chemotherapy; CRS, cytoreductive surgery; HIPEC, hyperthermic intraperitoneal chemotherapy; PCI, Peritoneal Cancer Index; PCI class: I, score 0–9; II, score 10–19; III, score 20–29; IV, score 30–39; NLR, neutrophil-to-lymphocyte ratio; M1, metastatic; OS, overall survival. ^a^. In this study, we had 2 and 3 patients diagnosed with the signet ring cell histotype, which was categorized into poor differentiation, in the curative- and the palliative-intent group, respectively. ^b^. A total of 15 patients in the curative-intent group received adjuvant therapy, and there were 4 cases of FLOT regimen (fluorouracil, leucovorin, oxaliplatin, and docetaxel), 2 cases of TC regimen (taxotere and cisplatin), 3 cases of target therapy (each had trastuzumab, pembrolizumab, and ramucirumab), 2 cases of capecitabine, and the last 4 cases receiving additional intraperitoneal infusions of taxotere.

**Table 3 cancers-15-02089-t003:** Univariate and multivariate analyses of preoperative factors on overall survival by Cox regression.

	Univariate ^a^			Multivariate		
	HR	95%CI	*p*-Value	HR	95%CI	*p*-Value
ECOG 0 1 2	1 1.48 6.09	0.60–3.65 2.05–18.06	0.395 0.001			
Preoperative NLR						
≤4.4	1			1		
>4.4	3.22	1.42–7.30	0.005	3.70	1.55–8.79	0.003
Histologic differentiation						
Well	1					
Moderate	2.57	0.93–7.09	0.068			
Poor	3.07	1.14–8.22	0.026			
Number of CT risks						
≤2	1			1		
>2	2.91	1.22–6.96	0.016	3.26	1.33–7.98	0.005

Abbreviation: HR, hazard ratio; CI, confidence interval; ECOG, Eastern Cooperative Oncology Group; CRS, cytoreductive surgery; NLR, neutrophil-to-lymphocyte ratio; CT, computed tomography. ^a^. The following preoperative prognostic factors were also calculated in the univariate analysis: age, gender, smoking, alcohol, diabetes, hypertension, abdominal operation history, pre-CRS CT, primary or recurrent cancer sources, severity of clinical symptoms, histologic features, number of CT risks; only significant results (*p* < 0.100) are shown in this table and evaluated in the multivariate analysis.

## Data Availability

The data presented in this study are available on request from the corresponding author. The data are not publicly available due to institutional restrictions.

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
