# Peer review of "Cytoreductive Surgery with Hyperthermic Intraperitoneal Chemotherapy for Gastric Cancer with Peritoneal Carcinomatosis: Additional Information Helps to Optimize Patient Selection before Surgery"

_cancers, 2023, doi:10.3390/cancers15072089_

Round 1
Reviewer 1 Report
The paper by Wu et al studies a short retrospective series of 44 patients to identify patient selection criteria for cytoreductive surgery and HIPEC in peritoneal carcinomatosis of gastric origin.
Title The title does not adequately reflect the results of the study and could be improved to better summarize the study's findings.
Abstract The abstract provides a good overview of the study and accurately reflects the results.
Introduction The introduction provides a concise and consistent background of the problem and clearly explains the rationale for the study.
Methods As this is a retrospective review of a specific time window, the number of patients studied is a result, and as such, it belongs in the Results section. The authors note that prophylactic HIPEC is given in non-metastatic gastric cancer patients at their center, which is not a standard treatment, and therefore the authors should explain the criteria used for prophylactic HIPEC and the full protocol for its administration. The authors could also provide more information on the role of laparoscopy in decision-making and could provide clarification on the administration of the NIPS protocol, including whether it is administered laparoscopically, after laparotomy with an intraperitoneal catheter, and if the PCI is calculated at that time. The authors should better define the neutrophil-to-lymphocyte ratio, including the reason for the cut-off of 4.4. Additionally, the authors should explain why PCI is not included in Table 3, which presents the univariate and multivariate analyses of preoperative factors on overall survival by Cox r.
Results The authors do not explain the variable CT prognostic risk, which is one of the primary variables of the study. The usefulness of Figure 2 is not entirely clear and could be better explained. The authors should provide more information on the PCI class, which is a variable that is introduced in the Results section but not in the Methods section. The usefulness of Figures 4a and 4b is unclear, especially given the significant differences in the surgical procedures performed between the curative and palliative groups.
Discussion The first paragraph of the Discussion section repeats information already presented, and could be made more clear and concise. The authors should compare their results to the current literature, as there is abundant research on PCI and peritoneal gastric cancer, as well as the neutrophil-to-lymphocyte ratio. The authors mention the signet cell ring, which is an important variable but do not include it in their study. The authors note the lack of consensus on PCI but could provide information on indications for a PCI <7, such as the findings of Manzanedo et al. (2019). Finally, since HIPEC is a locoregional therapy, the authors should include information on relapse-free survival, which could better illustrate the impact of HIPEC.
Conclusions The conclusions are somewhat vague, and do not fully address the stated aim of establishing preoperative patient selection criteria for CRS-HIPEC. The authors present risk factors, but could better summarize their findings and provide more concrete recommendations based on their results.
Author Response
1-1. Reviewer comment (Reviewer 1):
The paper by Wu et al studies a short retrospective series of 44 patients to identify patient selection criteria for cytoreductive surgery and HIPEC in peritoneal carcinomatosis of gastric origin. Abstract The abstract provides a good overview of the study and accurately reflects the results. Introduction The introduction provides a concise and consistent background of the problem and clearly explains the rationale for the study.
Title The title does not adequately reflect the results of the study and could be improved to better summarize the study's findings.
1-1. Author’s response:
We are grateful for these comments as positive affirmation. We thank the reviewer for this comment. There existed discrepancy between the results and the title. We modified the title to make it more relevant to the subject.
Title:
Cytoreductive surgery with hyperthermic intraperitoneal chemotherapy for gastric cancer with peritoneal carcinomatosis: Additional information helps to optimize patient selection before surgery
1-2. Reviewer comment (Reviewer 1):
Methods As this is a retrospective review of a specific time window, the number of patients studied is a result, and as such, it belongs in the Results section.
1-2. Author’s response:
We thank the reviewer for this comment. We modified the paragraph to the results section.
Page 4 Line 151:
A total of 73 consecutive CRS-HIPEC procedures for pathologically proven GC were reviewed. Patients with non-metastatic status who underwent prophylactic CRS-HIPEC operation (n=23) and those who underwent repetitive CRS-HIPEC procedures (n=6) were excluded. Subsequently, 44 GCPC cases were included in the study population, and 6 received neoadjuvant and intraperitoneal systemic chemotherapy (NIPS) before the formal CRS-HIPEC operation. All eligible patients were deemed to be medically fit for CRS-HIPEC procedures as follows: Eastern Cooperative Oncology Group (ECOG) performance ≤ 2 and absence of extra-abdominal metastasis. We categorized enrolled patients into the curative-intent (n= 20; as a therapeutic measure in those who achieved CCR 0-1) and the palliative-intent group (n=24; as symptomatic mitigation in those who attained CCR 2-3) according to the CRS completeness, as shown in Figure 1.
1-3. Reviewer comment (Reviewer 1):
The authors note that prophylactic HIPEC is given in non-metastatic gastric cancer patients at their center, which is not a standard treatment, and therefore the authors should explain the criteria used for prophylactic HIPEC and the full protocol for its administration. The authors could also provide more information on the role of laparoscopy in decision-making and could provide clarification on the administration of the NIPS protocol, including whether it is administered laparoscopically, after laparotomy with an intraperitoneal catheter, and if the PCI is calculated at that time.
Efficacy of conversion surgery after neoadjuvant intraperitoneal-systemic chemotherapy in treating peritoneal metastasis of gastric cancer
1-3. Author’s response:
To elaborate our institutional protocol and to help the reader understand more finely, we provide information with regards to prophylactic HIPEC and NIPS.
We wrote the following sentences in the materials and methods section.
Page 03 Line 95:
There is no strict protocol of prophylactic HIPEC for non-metastatic GC currently in our institute, and neoadjuvant intraperitoneal systemic chemotherapy (NIPS) for selected patients with evidence of presenting PC is also mainly introduced by physicians’ preference. Briefly, prophylactic HIPEC after curative gastrectomy has been considered an option as an adjuvant treatment to prevent peritoneal seeding or recurrence for high-risk patients (whose tumor invaded the visceral peritoneum of the stomach or the tumor directly invaded adjacent structures) as a previous study suggested [12]. NIPS is administered laparoscopically for selected advanced GC with evident PC, and the following intraoperative findings are recorded: the PCI score, PC location, and ascites amount. Intraoperative fluid cytology and biopsy are arranged according to clinical needs. The efficacy is evaluated after a full cycle of NIPS treatment (4 times), and early termination of NIPS is considered for conditions including rapid growing or newly developed PC lesions, or intractable post-procedure complications. Plateau response of GCPC to NIPS could lead to either curative- or palliative-intent CRS-HIPEC based on an individual’s clinical progression. Besides, we do not perform routine preoperative laparoscopic tumor staging and PCI evaluation before the CRS-HIPEC surgery.
1-4. Reviewer comment (Reviewer 1):
The authors should better define the neutrophil-to-lymphocyte ratio, including the reason for the cut-off of 4.4.
1-4. Author’s response:
We are grateful for this comment as it points to an important issue that should be elaborate more finely in the current study. We added the necessary changes in the methods section.
Page 03 Line 133:
Receiver operating characteristic (ROC) curve analyses were used to examine the the predictive value of clinical factors by the Youden index and to determine optimal cut-off points of numbers of CT risks and neutrophil-to-lymphocyte ratio (NLR).
1-5. Reviewer comment (Reviewer 1):
Additionally, the authors should explain why PCI is not included in Table 3, which presents the univariate and multivariate analyses of preoperative factors on overall survival by Cox r.
1-5. Author’s response:
We thank the reviewer for this comment and agree with the clinical significance of PCI score in Peritoneal surface malignancy patients. Generally, the PCI score is not a routinely available preoperative factor in all GCPC patients receiving CRS-HIPEC. Therefore, the PCI score was not entered into the Cox regression model. The two identified pre-operative prognostic factors are the foundation of the current study, our results showed that a combination emerging score of them was capable of predicting survival outcomes in GCPC recipients and provided additional value to PCI class, and may help optimize patients’ selection before CRS-HIPEC. Looking forward to the futurity, our findings may provide as additional information to evaluate GCPC patients before CRS-HIPEC.
We added the following sentences in the results section, also reuploaded the revised figure and manuscript.
Page 08 Line 235:
3.4. Additional pre-operative information to the PCI in clinical outcomes
The clinical significance of PCI score in Peritoneal surface malignancy patients contributing to clinical outcomes is evident but may not be always available pre-operatively. Therefore, we used ROC analyses to examine predictive values of the emerging combination of the two survival risks, identified from the Cox regression model assessing purely pre-operative factors and PCI class were examined, as shown in Figure 5. The patients were categorized into 3 groups by the number of independent survival risks a patient had (14 patients had 0 risk factors, 21 patients had at least 1 risk factor, and 9 patients had both risk factors). There were 10, 12, 16, and 6 patients in the PCI class I, II, III, and IV, respectively. The area under the ROC (AUROC) of the emerging numbers of survival risks in predicting the 6-month mortality, the 12-month mortality, and incomplete cytoreduction were 0.745 (95% CI: 0.597–0.893; Figure 5A), 0.737 (95% CI: 0.588–0.885; Figure 5B), and 0.632 (95% CI: 0.466–0.797; Figure 5C), respectively. On the contrary, predictive values of PCI class in the 6-month mortality, the 12-month mortality, and incomplete cytoreduction were 0.709 (95% CI: 0.556–0.863; Figure 5D), 0.675 (95% CI: 0.508–0.841; Figure 5E), and 0.786 (95% CI: 0.642–0.930; Figure 5F), respectively. Although the PCI class showed good result to predict incomplete cytoreduction but the discrimination ability was mediocre in prognosticating the 6-month and the 12-month survival outcomes. In addition, the emerging pre-operative risks scoring seemed to provide valuable information with regards to survival outcomes.
1-6. Reviewer comment (Reviewer 1):
Results The authors do not explain the variable CT prognostic risk, which is one of the primary variables of the study. The usefulness of Figure 2 is not entirely clear and could be better explained.
1-6. Author’s response:
We thank the reviewer for this comment. The significance of PCI score is evident contributing to CCR and CRS-HIPEC efficacy. In the present study, we aimed to discover pre-operative factors that have impact on survival after the surgery. Therefore, the association between these factors and PCI was investigated. We re-structured and added the following sentences in the Results section, also reuploaded the revised manuscript.
Page 04 Line 170:
The number of CT prognostic risks and GC primacy were discriminative in the curative-intent group. We had 8 (40%) and 17 (70.8%) patients with more than two CT risks in the curative- and palliative-intent groups, respectively (p = 0.040). We further investigated the relationship between the peritoneal cancer index (PCI) score and these two factors. There was a strong correlation between PCI score and CT risks (r = 0.534, p = 0.0001), whereas GC primacy did not correlate with PCI score (r = 0.153, p = 0.320), as shown in Figure 2.
1-7. Reviewer comment (Reviewer 1):
The usefulness of Figures 4a and 4b is unclear, especially given the significant differences in the surgical procedures performed between the curative and palliative groups.
1-7. Author’s response:
We are grateful for this comment. We made necessary explanations as followings and uploaded the revised Figure legend and manuscript.
Figure Legends, Figure 4:
Figure 4. Kaplan–Meier plots of overall survival according to different subgroups that contain with or without the identified survival risks. (A) Patients with lower preoperative serum NLR (≤ 4.4) demonstrated superior outcomes (p = 0.003), and (B) a more favorable outcome after CRS-HIPEC surgery was deliberated in patients with ≥ 2 CT prognostic risks (p = 0.011). NLR, neutrophil-to-lymphocyte ratio; CRS-HIPEC, cytoreductive surgery and hyperthermic intraperitoneal chemotherapy; CT, computed tomography.
1-8. Reviewer comment (Reviewer 1):
The authors should provide more information on the PCI class, which is a variable that is introduced in the Results section but not in the Methods section.
1-8. Author’s response:
We thank the reviewer for this comment. We made necessary explainations and add following sentences in the methods section to help the reader understand the PCI classification.
Page 02 Line 88:
The PCI score is a scoring system to measure the severity and extension of peritoneal surface malignancy based on divided intraperitoneal areas, point scores at each area are then added up and classified as: class I: 0-9 points, class II: 10-19 points, class III: 20-29 points, and class IV: 30-39 points.
1-9. Reviewer comment (Reviewer 1):
Discussion The first paragraph of the Discussion section repeats information already presented, and could be made more clear and concise. The authors should compare their results to the current literature, as there is abundant research on PCI and peritoneal gastric cancer, as well as the neutrophil-to-lymphocyte ratio. The authors mention the signet cell ring, which is an important variable but do not include it in their study. The authors note the lack of consensus on PCI but could provide information on indications for a PCI <7, such as the findings of Manzanedo et al. (2019). Finally, since HIPEC is a locoregional therapy, the authors should include information on relapse-free survival, which could better illustrate the impact of HIPEC.
1-9. Author’s response:
We thank the reviewer for this comment. We have rewritten the first paragraph of the Discussions section and compare our results regarding the PCI and NLR with previous studies to help the reader understand our main messages. Information of relapse-free survival was also provided.
Page 08 Line 257:
This research aims to try to identify the potential GCPC beneficiaries from CRS-HIPEC operation by analyzing the relationship between the clinicopathological information and survival outcomes for GCPC after CRS-HIPEC. In this present study, the 3-year survival rate in the curative-intent group was 38.6%. Moreover, relapse-free survival (RFS) was also assessed in patients who achieved CCR 0 (n=12), and the median RFS was 18.2 months (mean ± SD: 20.7±5.2; 95% CI: 10.4-30.9). Our patients with PCI class I had a median OS of 28.2 months (OS at 1-, 2- and 3-year of 88.9%, 74.1%, and 37.0%), while those with PCI class II, III, and IV had a median OS of 10.0, 5.7, and 2.2 months, respectively. In Manzanedo’s series with a PCI cut-off point of 7, their patients with PCI < 7 had a median OS 26.1 months [17]. Notably, two GCPC patients with PCI scores of 3 and 10, attained long-term disease-free survival of over 50 months after CRS-HIPEC surgery in our series. To date, there is still no current consensus on an optimal PCI score cut-off value for patients with GCPC to receive CRS-HIPEC [17,18]. A higher PCI score is considered to be correlated with a heavier tumor burden, which increases the difficulty in achieving complete cytoreduction. When a PCI score exceeded 13, the rate of completeness of cytoreduction was only about 7% [19], and we discovered that only 7 of 29 (24.1%) patients achieved CCR 0-1 in the present study… …
We totally agree with the significance of the signet ring cell histotype, however, it was not identified as an independent risk and has no proportional difference between the two groups. We added footnote b to illustrate in the Table 2:
‘’ In this study, we had 2 and 3 patients diagnosed as the signet ring cell histotype, which was categorized into the poor differentiation, in the curative- and the palliative-intent group, respectively.’’
1-10. Reviewer comment (Reviewer 1):
Conclusions The conclusions are somewhat vague, and do not fully address the stated aim of establishing preoperative patient selection criteria for CRS-HIPEC. The authors present risk factors, but could better summarize their findings and provide more concrete recommendations based on their results.
1-10. Author’s response:
We thank the reviewer for this comment. We have rewritten the conclusions to help the reader understand our main message and uploaded the revised manuscript.
Conclusions:
In the current study, we found two easily measurable and promising pre-operative risk factors (number of CT prognostic risks > 2 and serum NLR > 4.4) that can be used to select ideal candidates for CRS-HIPEC operation and to predict post-surgical survival benefits. Our findings not only provide essential values, especially for those who lack or have difficulty in assessing the exact PCI score but also give information in addition to the PCI score. Looking forward to the future, these results should inspire developments in optimizing GCPC patient selection prior to the CRS-HIPEC operation.

Reviewer 2 Report
This is a nicely written paper. The authors already included the main limitations in the discussion section: retrospective design and small sample size.
A central aspect of this manuscript is "NLR". The NLR is not tumor specific and has been shown to correlate with tumor burden in many gastrointestinal tumors. It is therefore not fair to put so much importance on this finding because peritoneal metastasis represents advance tumor mit high tumor burden (as proven in this series). It would have been interesting to investigate a possible correlation between NLR and tumor differentiation.
Nonetheless, I would like to congratulate the authors for this good paper.
Author Response
2-1. Reviewer comment (Reviewer 2):
This is a nicely written paper. The authors already included the main limitations in the discussion section: retrospective design and small sample size.
A central aspect of this manuscript is "NLR". The NLR is not tumor specific and has been shown to correlate with tumor burden in many gastrointestinal tumors. It is therefore not fair to put so much importance on this finding because peritoneal metastasis represents advance tumor mit high tumor burden (as proven in this series). It would have been interesting to investigate a possible correlation between NLR and tumor differentiation.
Nonetheless, I would like to congratulate the authors for this good paper.
2-1. Author’s response:
We are grateful for this comment as it points to an important issue that should be discussed in the future study. There existed distinct proportions in the histology in the current study that most of the enrolled cases, 38 of 44 (86.4%), were diagnosed with a poorly differentiated histotype. For your information, details were accessed according to different histotypes and NLR, and there were 1(4.0%) well, 2 (8.0%) moderately and 22 (88.0%) poorly-differentiated histotypes with a low NLR and 3(15.8%) well, 0 (0.0%) moderately and 16 (84.2%) poorly differentiated histotypes with a high NLR. The distribution was not statistically different (p-value = 0.203).
We also adjusted the part of NLR in balance relation to a whole in the discussion section.
Reviewer 3 Report
Thank you for the opportunity to review this manuscript.
Gastric cancer with peritoneal carcinomatosis is a disease with a very poor prognosis and there are only a limited opportunities for control of the course of the disease.
In gastric cancer, patient selection for CRS and HIPEC is crucial.
The main limitation of this study is the relatively very small group of patients with radical surgery (cohort of 20 patients).
In the trial group, the importance of adjuvant chemotherapy (FLOT regimen?) or targeted immunotherapy is unclear.
It should be better described.
Author Response
3-1. Reviewer comment (Reviewer 3):
Gastric cancer with peritoneal carcinomatosis is a disease with a very poor prognosis and there are only a limited opportunities for control of the course of the disease.
In gastric cancer, patient selection for CRS and HIPEC is crucial.
The main limitation of this study is the relatively very small group of patients with radical surgery (cohort of 20 patients).
In the trial group, the importance of adjuvant chemotherapy (FLOT regimen?) or targeted immunotherapy is unclear.
It should be better described.
3-1. Author’s response:
We thank the reviewer for this comment. We totally agree with the significance of adjuvant therapy in survival. We added footnote c to illustrate in the Table 2:
‘’ Fifteen patients in the curative-intent group received adjuvant therapy, and there were 4 cases of FLOT regimen (fluorouracil, leucovorin, oxaliplatin, and docetaxel), 2 cases of TC regimen (Taxotere and cisplatin), 3 cases of target therapy (each had trastuzumab, pembrolizumab, and ramucirumab), 2 cases of capecitabine, and the last 4 cases receiving additional intraperitoneal infusion of taxotere.’’
Round 2
Reviewer 3 Report
In my opinion, the manuscript is acceptable for publication in this form.